# Tissue-Specific Proteome and Subcellular Microscopic Analyses Reveal the Effect of High Salt Concentration on Actin Cytoskeleton and Vacuolization in Aleurone Cells during Early Germination of Barley

**DOI:** 10.3390/ijms22179642

**Published:** 2021-09-06

**Authors:** Georgi Dermendjiev, Madeleine Schnurer, Jakob Weiszmann, Sarah Wilfinger, Emanuel Ott, Claudia Gebert, Wolfram Weckwerth, Verena Ibl

**Affiliations:** 1Department of Functional and Evolutionary Ecology, Molecular Systems Biology (MoSys), Faculty of Life Sciences, University of Vienna, Djerassiplatz 1, 1030 Wien, Austria; peroxysom@gmail.com (G.D.); madeleine.schnurer@univie.ac.at (M.S.); j.weiszmann@gmail.com (J.W.); a01620984@unet.univie.ac.at (S.W.); emanuel.ott@gmx.at (E.O.); c.gebert94@gmx.at (C.G.); wolfram.weckwerth@univie.ac.at (W.W.); 2Vienna Metabolomics Center (VIME), University of Vienna, Djerassiplatz 1, 1030 Wien, Austria

**Keywords:** barley, germination, mobilization, vacuolization, proteomics, salt stress

## Abstract

Cereal grain germination provides the basis for crop production and requires a tissue-specific interplay between the embryo and endosperm during heterotrophic germination involving signalling, protein secretion, and nutrient uptake until autotrophic growth is possible. High salt concentrations in soil are one of the most severe constraints limiting the germination of crop plants, affecting the metabolism and redox status within the tissues of germinating seed. However, little is known about the effect of salt on seed storage protein mobilization, the endomembrane system, and protein trafficking within and between these tissues. Here, we used mass spectrometry analyses to investigate the protein dynamics of the embryo and endosperm of barley (*Hordeum vulgare*, L.) at five different early points during germination (0, 12, 24, 48, and 72 h after imbibition) in germinated grains subjected to salt stress. The expression of proteins in the embryo as well as in the endosperm was temporally regulated. Seed storage proteins (SSPs), peptidases, and starch-digesting enzymes were affected by salt. Additionally, microscopic analyses revealed an altered assembly of actin bundles and morphology of protein storage vacuoles (PSVs) in the aleurone layer. Our results suggest that besides the salt-induced protein expression, intracellular trafficking and actin cytoskeleton assembly are responsible for germination delay under salt stress conditions.

## 1. Introduction

Since seed germination is the most crucial phase for plant growth and development and subsequently influences crop yield and quality, germination studies of crop plants are an important aspect of plant biology [1].

The cereal germination process starts with the uptake of water by the quiescent dry grain (imbibition), triggering the degradation of food reserves [2,3]. Three phases are characterized within this process [4]: Phase I, the early phase, where the imbibition of the dry grain takes place, and an early plateau phase of the water uptake is reached. Phase II, the middle phase, which ends with the visible radicle protrusion through the coleorhiza; and phase III, the late phase, which encompasses subsequent seedling development.

Cereal grains contain distinct layers, each with different spatiotemporal physiological roles and molecular mechanisms [5,6]: the endosperm, the major seed storage tissue; the dry coat on the outside, which includes the pericarp/fruit coat; the seed coat, which encloses the embryo; and the scutellum in the embryo, which stores high amounts of lipids and proteins. Large amounts of seed storage proteins (SSPs) and starch are stored within the endosperm [7,8]. Nowadays, SSPs are classified based on functional or structural and evolutionary relationships: (i) the CUPIN superfamily (globulins) and (ii) the PROLAMIN superfamily (prolamins, α-amylase/trypsin inhibitors, non-specific lipid transfer proteins, 2S albumins, puroindolines (PINs), hydrophobic protein, α-globulines, and hydroxyproline-rich proteins) [9]. During cereal grain germination, stored lipids, carbohydrates, and proteins are mobilized to the embryo. For the activation of this mobilization process, gibberellic acids (GA_3_) accumulated in the embryo are transported to the aleurone layer of the endosperm, a living tissue surrounding the dead starchy endosperm. Hydrolytic enzymes, including acidic cysteine endopeptidase, serine carboxypeptidases, and neutral aminopeptidases, are activated to break down the stored protein reserves in the endosperm [10,11]. Additionally, expression and secretion of α-amylase by the aleurone and the scutellum are induced to start the breakdown of starch in the starchy endosperm [12].

For a successful germination, the environmental parameters, such as sufficient moisture and optimal temperature, are key players. Further, soil salinity has a large impact on the yield of a wide variety of crops all over the world, both inhibiting plant growth and affecting the germination as well as root and shoot development through osmotic stress and ion toxicity [13]. In Europe, for instance, salinization occurs by the accumulation of salts from irrigation water and fertilizers, leading to the rapid accumulation of soluble salts in soil. Overall, natural and artificial soil salinization is observed in around 3.8 million hectares in Europe [14,15].

Plants can be classified into salt—tolerant halophytes and salt—intolerant glycophytes based on their response to salt stress [16]. Besides altering farming practices to prevent soil salinization by implementing programs to remediate salinized soils, traditional breeding or genetic manipulation technologies have been implemented to increase the salt tolerance of plants.

Barley, which is the fourth most important crop in terms of food production after maize, wheat, and rice (FAOSTAT 2016), is considered as a marginal halophyte as it can tolerate between 150 mM and 500 mM NaCl [17,18]. To understand the role and function of protein mobilization of early germinated barley during the salt stress response, insights into the proteome remodelling process of germination will help us to understand the germination process under altered environmental conditions. Proteome and metabolome changes of the barley grain, the embryo, the endosperm, as well as changes in the redox state and in the expression of defence-related proteins have been identified using germinated grains, the aleurone layer treated with GA_3_, and germinated embryos on agar plates in normal and salt-stressed conditions [19,20,21,22,23].

With this view in mind, we investigated the protein dynamics of the embryo and endosperm in germinated grains over time to obtain cellular adaptations in barley grain germinating under high salt conditions. We analysed the proteome of the embryo and the endosperm at five different early points during germination (0, 12, 24, 48, and 72 h after imbibition (HAI)) and first (i) identified a temporally regulated change in the abundance of proteins, including SSPs, peptidases, and starch hydrolyses, and (ii) determined the affected proteins that were involved in the protein trafficking or cytoskeleton-related proteins. Microscopic analysis of a transgenic line that visualizes the vacuole and use of a staining dye to visualize the actin cytoskeleton confirmed that actin bundling, and vacuolar morphology were changed under salt stress conditions in aleurone cells.

## 2. Results

### 2.1. Application of Natural Conditions to the Lab: First Steps to #Asnearaspossibletonature

As temperature has an effect on the germination rate, growth rate, and total protein content during barley germination [24], we tried to use a suitable temperature representative of central Europe for barley early germination processes. To obtain this temperature, we set up a field trial in 2019, where we followed the germination and growth of the wild-type barley cultivar Golden Promise (GP). The seeds were sown on the 3 April 2019 (Figure 1a) and shoots could be observed on the 11 of April 2019 (Figure 1b). The lowest temperature during this germination process ranged between 10 °C and 15 °C (Figure 1c). Interestingly, the soil accumulated heat, and the highest temperatures were seen after nightfall, with the temperature decreasing during the late hours of the night and increasing again during the day (Figure 1c).

### 2.2. The Germination Rate of GP Is Reduced under High Salt Conditions (EC30) Compared to Tap Water (H_2_O)

We used the obtained data from our field experiment to provide the environmental conditions of our controlled experiment, i.e., a 14 °C/12 °C day/night cycle. To assess the physiological as well as molecular changes in the embryo and the endosperm in germinated grains, barley GP grains were germinated in tap water treatment (hereinafter abbreviated as H_2_O) and in tap water with added NaCl (electric conductivity (EC) 30 mS/cm; treatment hereinafter termed EC30) in Petri dishes. The germination processes of the control samples as well as the treated samples were followed specifically at 12, 24, 48, and 72 HAI, where seeds were physiologically analysed as to their appearance and length of the radicle and coleoptile measured and dry weight and wet weight assessed as well. Here, we focused on the germination *sensu stricto*, i.e., phase I and II, from the start of imbibition of the dry seed until when the radicle first emerged [25]. Thus, considering a grain as germinated when the radicle was 1 mm long, most grains were germinated after 48 HAI in the H_2_O treatment but only a few at 72 HAI in EC30 (Figure 2a). Viability staining using 1% TCC (2,3,5-Triphenyltetrazolium chloride (TTC) of the germinated grains showed that the embryo was metabolically active during EC30 treatment (Figure 2b), but salt treatment caused a decrease of 90% in the germination rate (Figure 2c). Additionally, we measured the starting, wet, and dry weight to assess the water uptake until 72 HAI. While the grain weight increased around 7 times in GP grains soaked in H_2_O compared to the starting and dry weight, the weight of GP grains incubated in EC30 only increased about 4 times (Figure 2d), indicating that less water was taken up when GP grains were subjected to salt stress.

### 2.3. Characterization of the Proteome of Early Germinated Barley Grains in H_2_O Versus EC30 Treatments

To investigate the effect of salt at the molecular level in the embryo and in the endosperm, we investigated the protein dynamics of the embryo and endosperm from grains that were germinated in the H_2_O and in EC30 treatments. Grains were harvested for protein extraction at 0, 12, 24, 48, and 72 HAI as follows: grains were cut longitudinally, and the embryo was separated from the endosperm and aleurone by an additional transverse section and finally removed by tweezers from both halves of the grain. Protein extraction was done from the embryo and endosperm at all time points.

We identified and quantified in total 2066 proteins within the embryo and 1090 proteins within the endosperm of grains germinated in H_2_O, and 2088 proteins within the embryo and 1162 proteins within the endosperm of grains germinated in the EC30 treatment (Appendix A). Interestingly, the number of unique identified proteins between either treatment (H_2_O or EC30) increased between 12 and 72 HAI (Appendix A).

Unsupervised multivariate statistical analysis clustered the endosperm and embryo stages separately, as indicated by principal component analysis (PCA) (Figure 3). PCA analysis (34.07% of PC1 variance) of the different germination stages of the embryo revealed the separation of four clusters (Figure 3a). This was also true for the endosperm, where the investigation of the PCA results showed that PC1 (18.17% of PC1 variance) of the proteome of the endosperm separated the samples of the different points of time within four clusters (Figure 3b). A specific change of the proteome could be observed at 48 and 72 HAI between H_2_O and EC30 in the embryo as well as in the endosperm (Figure 3a,b).

To understand the effect of the EC30 treatment specifically on the proteome of the embryo and endosperm, the protein abundance was analysed during the germination of each tissue. Within this analysis, we identified and quantified in total 1130 proteins, with 375 proteins found in the endosperm and 755 proteins in the embryo (Appendix A). Hierarchical bi-clustering analysis (HCA) clustered the 755 proteins of the embryo and the 375 proteins of the endosperm, respectively, in more abundant (*) and less abundant (**) clusters at the early stages of the germination process (Figure 4a,b). Additionally, we performed a two-way ANOVA analysis that identified significant differences within the H_2_O and EC30 treatment, between the H_2_O and EC30 treatments, and comprising all treatments (Appendix A). Proteins that were significantly differently expressed between H_2_O and EC30 at specific stages were clustered according to the Gene Ontology and GO annotation (https://www.ebi.ac.uk/QuickGO/, accessed on 21 May 2021) into the following groups: “cellular”, “metabolic”—including proteins involved in the anabolism pathway, “protein/lipid transport”, “catalytic”, “catabolic”, “unknown”, “defence”, and “seed storage protein (SSP)” (Figure 4c,d).

Given the dependency of the signalling pathways between the embryo and the endosperm, we asked whether the response to EC30 is temporally regulated in the embryo and the endosperm. We assayed the number of the differently expressed proteins (DEPs) in the different stages of the embryo as well as in the endosperm. We found 339 DEPs at 48 HAI followed by 218 at 72 HAI in the embryo. Interestingly, 122 proteins were differently abundant at 72 HAI followed by 61 proteins at 48 HAI in the endosperm (Appendix A). Thus, according to the number of differently expressed proteins, the embryo reacted first to the EC30 conditions at 48 HAI and the endosperm followed at 72 HAI.

### 2.4. The Energy Resource Mobilization during the Germination Process

During the heterotrophic germination process, SSPs (including proteins from the CUPIN superfamily (globulins) and the PROLAMIN superfamily increase in abundance [9]. As we detected a delay in the germination process when mature grains were subjected to EC30, we asked if the abundance of distinct SSPs is different in the embryo and in the endosperm when germinated in H_2_O versus EC30 conditions. Indeed, we could identify in total 19 differently regulated SSPs between treatments in the embryo and the endosperm (Figure 5, Appendix A). We found seven SSPs only from the CUPIN family in the embryo that were more abundant at 48 and 72 HAI in H_2_O compared to EC30 (Figure 5a). Moreover, we could identify 20 peptidases that were significantly abundant in EC30 embryos (Figure 5a). In the endosperm, we observed that most of the SSPs were more abundant when germinated in EC30 (Figure 5b). Using HCA analysis, two clusters were observed, which showed that the SSPs of the PROLAMIN family were most abundant at 72 HAI when subjected to EC30 (Figure 5c). The strong abundance of the peptidases at 72 HAI in the endosperm in H_2_O (Figure 5b) compared to EC30 indicated that the SSPs of the PROLAMIN group are not digested in the endosperm when subjected to salt treatment.

Given that the combined action of α-amylase, β-amylase, α-glucosidase (maltase), and limit dextrinase is necessary for the hydrolysis of starch in the endosperm, we expect these proteins to be present in the endosperm during germination [26]. Notably, these enzymes are produced in the aleurone (and in the scutellum) except for β-amylase, which is synthesized and stored during grain development in the starchy endosperm [27,28]. We identified and quantified α-amylase (A0A287UTQ8), α-glucosidase (M0Y8T5) in both tissues, and *β*-amylase and limit dextrinase specifically in the endosperm (Figure 6a, Appendix A). Within the embryo, α-amylase and α-glucosidase were significantly more abundant in H_2_O than EC30 at 48 HAI. In the endosperm, the protein abundance of α-amylase, α-Glucosidase, and limit dextrinase (A0A287VUA9) was significantly higher at 48 and 72 HAI when subjected to salt stress whereas the abundance of *β*-amylase showed no difference (Figure 6a, Appendix A).

Recent transcriptional and microscopic analyses showed that barley endo-β-mannanase 1 (HvMAN1) from the Hordeum vulgare MAN gene family (HvMAN1, HvMAN3, HvMAN4, and HvMAN6) is synthesized in the aleurone layer and then moves through the apoplast to the starchy endosperm to digest the mannan polymers in the cell walls, which can be used as energy resources [29]. Our investigation of protein abundances of the endosperm identified HvMAN4 (A0A287E459) at 48 and 72 HAI only in H_2_O but not in the EC30 treatment (Figure 6b, Appendix A).

These results show that salt treatment alters the protein abundance of starch and cell wall hydrolysing proteins in the endosperm at 48 HAI.

### 2.5. The Actin and Tubulin Cytoskeletons and the Rearrangement of Vacuoles Are Affected by EC30 in the Endosperm

Given the secretion system of the aleurone starts the mobilization of storage proteins during germination, a dynamic endomembrane system could be expected in this tissue. Microscopic analysis showed that protein storage vacuoles (PSVs) in the aleurone undergo morphological reshaping during germination [30]. These events include actin-dependent expansions of tubular structures [30]. Indeed, we detected five cytoskeleton-related proteins at 48 and 72 HAI with an altered protein abundance between H_2_O and EC30 in the endosperm (Figure 7a, Appendix A). Live-cell imaging analysis of germinated barley grains at 72 HAI using CellMask^TM^ Green Actin Tracking Stain showed an altered morphology of the actin cytoskeleton in the aleurone cells in the EC30 treatment (Figure 7b). Moreover, PSV tubules were detected during time-series (*t* = 5 min) when germinated in H_2_O but not in EC30 (Figure 7b). Since the spatial organization of protein trafficking, including all aspects of the endomembrane organelles and vesicular trafficking, is closely correlated to a correct function of the cytoskeleton [31], we analysed our data in terms of the proteins involved in vesicular trafficking. The most altered protein abundance was identified at 72 HAI followed by 48 HAI. We next analysed the biological processes at these points in time. Indeed, proteins involved in the protein/lipid transport appeared at 48 and 72 HAI in the endosperm (Appendix A). Specifically, four proteins involved in protein/lipid trafficking were altered in terms of their expression in the endosperm between H_2_O and EC30 (Figure 7c). Additionally, the abundance of proteins involved in cellular processes was altered at 48 HAI and 72 HAI (Appendix A). These data are consistent with a recent study showing that proteins involved in vesicular trafficking are affected in salt stress during hulless barley germination [32].

Altogether, these analyses show that the abundance of proteins involved in protein trafficking as well as the cytoskeleton-related proteins was altered when subjected to EC30. Additionally, we observed an increase in the extent of actin filament bundling as well as a reduction of dynamic rearrangements of PSVs by EC30.

## 3. Discussion

### 3.1. Getting Away from Petri Dishes to More Natural but Controlled Settings in the Lab: #Asnearaspossibletonature

Plant performance is strongly affected by environmental conditions. Subsequently, the results of controlled lab conditions—“pampered inside, pestered outside”—often do not translate back to field conditions: “Besides phenotypical differences between lab- and field-grown plants, the shoot and root environment and the effects of plant density must be considered” [33]. Recently, we successfully set up conditions to follow the germination, early root development, and grain filling processes at conditions as near as possible to nature (#asnearaspossibletonature). Light levels as well as soil and air temperature were measured between the 3 April 2019 and 11 April 2019. We set up lab conditions for germination according to these results. Of course, collecting data for several years is indispensable to receive a more complete picture of the range of conditions in the area during germination. Indeed, we continued the collection of data in a field trial of a bio-organic farmer in lower Austria for spring barley in 2021, supporting our germination temperature of around 14 °C. Subsequently, the next steps will be to follow the germination #asnearaspossibletonature in rhizoboxes directly in the soil under controlled environmental settings.

### 3.2. 48 HAI Could Be a Key Point in Time for the Germination under EC30-Like Conditions

PCA showed a clear separation of protein abundance in the embryo between 0 and 12 HAI and 24 HAI when germinated in H_2_O (Figure 3). Within the endosperm, a clear separation of protein abundance was observed between 0, 48, and 72 HAI (Figure 3). Within the embryo, most DEPs between germination in H_2_O and salt were found at 48 HAI followed by 72 HAI, whereas in the endosperm, most of the differently expressed proteins were found at 72 HAI, followed by 48 HAI (Appendix A). According to the gene ontology clustering, proteins with altered expression between the H_2_O and EC30 treatment were involved in metabolic pathways, in cellular activity, and SSPs, indicating the first vast anabolism response and the start of the synthesis of proteins and amino acids are affected by salt. This is clearly supported by the increase in the germination rate and the grain weight at this point in time when germinated in H_2_O.

### 3.3. The Antagonistic Regulation of GA_3_ and ABA Is Putatively Affected by Salt

GA_3_ and ABA antagonistically regulate the developmental transition from embryogenesis to seed germination [34]. In detail, GA_3_ induces the expression of α-amylase and hydrolase, and ABA inhibits these expressions by the expression of ABA-induced protein kinase [35]. Interestingly, we did not identify any GA_3_-responsive proteins in the embryo and in the endosperm (Appendix A). Salt stress triggers the accumulation of ABA in plant tissues and induces the expression of a number of ABA-induced proteins and water deficit-responsive genes [36]. We detected two proteins responding to abscisic acid (A0A287FU82, Em protein H5; A0A287FU93, Em protein CS41) that were only expressed in EC30 at 24 HAI and in 48 HAI, respectively (Appendix A), indicating induction of the ABA-signalling pathway in EC30. Additionally, desiccation-related LEA (late embryogenesis abundant) proteins, including dehydrins, that were significantly more abundant in the embryo at 12, 24, and 48 HAI when subjected to salt stress are known to be induced by ABA [37] (Appendix A).

### 3.4. Seed Storage Proteins in the Endosperm Are Less Digested under EC30 Conditions

A hallmark of the germination of cereal grains is the interplay between the embryo and the endosperm. SSPs must be hydrolysed by proteases to sustain embryo growth and development by delivering amino acids and sugar to the embryo until autotrophic growth is reached. In monocots, around 80% of cereal grain proteins are storage proteins, where cupin-like as well as prolamin SSPs are the major parts of the energy reserves in barley endosperm [38,39]. Barley prolamins are termed hordeins and account for >50% of the total protein amount in mature seeds [40,41]. Barley hordoindolines (HINs), HINa, HINb1, and HINb2, are orthologous proteins of wheat PINs and are small, basic, cysteine-rich, and seed-specific proteins that are responsible for grain hardness, a major quality trait [42]. PINs show a close structural relationship to 2S storage proteins and were proposed to be 2S-like storage proteins that may interact with prolamins via their tryptophan-rich domain [43]. These proteins are spatiotemporally synthesized and stored during barley endosperm development [44,45,46] and finally mobilized and degraded during germination. Among the proteases involved in the germination process, cysteine proteases (CysProt) from the papain family (C1A) and the legumain family (C13) are responsible for 90% of the proteolytic activity [47]. Additionally, members of the S10 serine carboxypeptidases (SCPs) and Papain-like CysProt participate in different stages of the development in rice, triticale, and *Brachypodium distachyon* (reviewed in [48]).

In both embryo and endosperm tissues, the degree of the digestion of SSPs increased with the developmental stage during germination in H_2_O. This is in line with transcriptomic data that describes a decline of the transcripts of SSPs and an increase of the proteases in the embryo and in the endosperm during barley germination [49]. However, the imbibition in EC30 delays this digestion, which is strongly supported by the high abundance of SSPs at 48 HAI and 72 HAI in EC30 in the embryo as well as in the endosperm and by the increase of peptidases at 72 HAI in EC30 compared to H_2_O. In parallel with the expression and secretion of peptidase, GA_3_ and ABA antagonistically regulate the production of α-amylase: the abundance of α-amylase mRNA increases in response to GA_3_ and is inhibited by ABA [50]. Additionally, GA_3_ induces cell wall digestion in the aleurone [51]. We detected and quantified the non-storage polysaccharide degradation enzyme manninose and the ß-Glucan degradation protein ß-D-Glucan glucohydrolase ExoI (A0A287SCK9), which were significantly higher in the endosperm when germinated in H_2_O at 48 and 72 HAI. (Appendix A). It should be noted that many so far published experiments regarding the spatial and molecular identification of proteases were performed either during malting processes [52], or grains were cut into halves or the aleurone layer was isolated and germination was induced by the usage of GA_3_ [53,54,55,56,57,58,59]. As we used germinated grains as our sample material, the absence of certain expected proteins is likely due to their low abundance in our defined experimental setting.

It has further been suggested that the embryo plays a major role during grain germination as new biosynthesis of proteins in the endosperm is putatively regulated by the embryo. The increase of the protein abundance of distinct proteins enables the degradation of reserves in the endosperm [60]. This is an interesting model and may explain the increase of the protein abundance of few SSPs in the endosperm (Figure 5b).

### 3.5. The Cytoskeleton Is Affected by the EC30 Treatment during Germination

Salt induces a complex cytoskeletal regulatory network [61]. Salt stress-induced cytoskeleton dynamic changes include the actin filament assembly and depolymerization as well as the microtubule depolymerization [61,62]. Our findings revealed that the expression of cytoskeleton-associated proteins was altered in the embryo (Appendix A) as well as in the endosperm when subjected to the EC30 treatment (Figure 7a). Interestingly, the actin key regulator cyclase-associated protein (CASP) [63,64] was only found in the embryo (Appendix A). Future tissue-specific studies within the embryo could reveal the specific function of the actin cytoskeleton in developing barley embryo, which is dynamically organized to ensure proper outgrowth of the seedling, probably by enhancing cell elongation. However, we could detect an altered morphology of actin bundling in the aleurone cells (Figure 7b) in the EC30 treatment, underlining the proteomics data showing that salt stress has an effect on cytoskeleton assembly.

### 3.6. Vacuolization Is Inhibited by EC30

The actin inhibitor latrunculin b inhibits the formation of PSV tubules in germinating barley grains [30]. Subsequently, we raised the question of whether the altered expression of cytoskeleton-related proteins and the affected actin influence the dynamic behaviour of PSV in the aleurone, as flexibility of reshaping vacuolar morphology is crucial for cell expansion and together with constantly adapting actin filament bundling drives proper plant growth [65]. The role of these proteins and actin was reflected by the reduced growth in EC30 (Figure 2a,b and Figure 7b).

### 3.7. Actin Is a Putative Key Player Not Only for Vacuolization but for the Transport of Secretory Proteins from the Aleurone to the Endosperm

Most of the peptidases and hydrolytic enzymes seen in this study are synthesized in the aleurone and secreted into the starchy endosperm over the secretory pathway [28]. After fusion of secretory vesicles with the plasma membrane, their content is released to the exterior of the cell [66] followed by secretion facilitated by cell wall channels and plasmodesmata to the outside of the aleurone cell wall [59]. Since the storage endosperm dies during seed development, any increase in the transcript or protein abundance must be due to newly synthesized RNA/protein in the aleurone that is added by secretion to the already stored proteins in the storage endosperms [20]. The aleurone is not only secreting hydrolytic enzymes, peptidases, and cell wall-degrading enzymes, but also proteins involved in the defence responses [28]. Our results showed that most of the secreted defence proteins were less abundant in EC30. Since the activation of reserve mobilization already starts at 24 HAI [20], it is not surprising that most of the differences in the protein abundance appeared at 48 and 72 HAI in the endosperm.

HVA22 is an ABA-induced LEA protein that inhibits vesicular trafficking during vacuolation in the aleurone, which is GA_3_ mediated [67]. We not only detected several ABA-induced LEA proteins but several protein trafficking-related proteins whose expression was altered within the embryo as well as in the endosperm when germinated in EC30 (Appendix A). Notably, one coatomer subunit was significantly downregulated in the embryo when the grains were subjected to EC30; we suggest that these proteins putatively negatively affect the ER-Golgi trafficking within the embryo. Interestingly, the expression of phospholipase D α1, which is described as being involved in salt tolerance in wheat and rice [68,69], was not significantly altered in response to EC30 in the embryo (M0 × 1N3) and the endosperm (A0A287KG55) (Appendix A) but expressed in H_2_O at 72 HAI. However, the proteins of the PLD family are expressed in different tissues, concomitant with distinct functions. Thus, PLDα1 putatively functions in ABA suppressed and GA_3_-stimulated α-amylase gene expression, as it has been described in barley aleurone cells [35,70,71].

The interpretation of our results is summarized in our proposed model (Figure 8). As the abundance of many proteins, including hydrolytic enzymes, are regulated at the transcriptional level in the aleurone and a smaller part in the embryo [49], we hypothesize that a lower abundance of proteins in the endosperm is related to lower transcriptional activity for these proteins in the aleurone. However, we cannot rule out any effect on the secretory pathway, as the expression and morphology of cytoskeleton-associated proteins, as well as proteins involved in protein trafficking and the reshaping of the vacuole were affected by the salt treatment.

## 4. Materials and Methods

### 4.1. Soil Temperature Data

To identify and accumulate data on soil temperature during typical barley germination conditions, we set up a field experiment at the University of Vienna, Althanstrasse 14. A total of 5000 seeds of the barley cultivar Golden Promise (GP) were sown in an approximately 15 m^2^ field on the 3 April 2019, and shoots could be observed on the 11 April 2019. Tensiomark sensors (ecoTech, Bonn, Germany) were used to constantly measure the soil temperature at a soil depth of −20 cm at four different randomized places to increase the reliability of the measured parameters. Measurements were conducted every 15 min and recorded by independent enviLog recording stations (ecoTech, Bonn, Germany) from the 3 April until the 4 August 2019.

### 4.2. Germination and Viability Assay

Experiments were performed in growth chambers at a 14 °C/12 °C day/night cycle with 70% humidity, in complete darkness. For germination assays, Petri dishes of Ø 9.5 cm diameter with filter paper, filled with either 5 mL of H_2_O (tap water, EC 0.4 mS/cm)) or EC 30 mS/cm (2 g NaCl in 100 mL tap water), were used. A total of 10 seeds were placed in each dish to assess imbibition/germination. Dishes were covered with aluminium foil to prevent any effects of light on the germination processes. A total of 10 technical replicates (seeds) per petri dish and 3–24 biological replicates (dishes) per sample group were used. Seeds were counted as germinated that showed the radicle (1 mm), according to [72]. In addition, pictures of the seeds were taken for documentation. To observe the physiological effects of salt on the viability of the embryo development, seeds were cut longitudinally with a razor blade and embryos were stained with 1% TCC (2,3,5-triphenyltetrazolium chloride (TTC) (VWR, Radnor, PA, USA) for 30 min to 1 h at 37 °C. For a negative staining control, preheated seeds (100 °C, 1 h) were stained with TTC, and no staining was observed.

### 4.3. Measurement of Weight, Wet, and Dry Weight of GP Grains during Germination

GP seeds (10 each) of 6 biological replicates were weighed before imbibition (starting weight) and after 12, 24, 48, 72, and 96 h after imbibition (HAI) in tap water or EC30 solution (wet weight). After measuring the wet weight, the seeds were put in a drying cabinet at 60 °C for 48 h. Subsequently, the dried seeds were weighed again (dry weight). The data were analysed using the program GraphPad Prism version 9.0 for Mac, GraphPad Software, San Diego, CA, USA.

### 4.4. Protein Extraction and Mass Spectrometry-Analysis

Total proteins were extracted from the endosperm as well as from embryos of 10 barley grains at the mature stage, and at 12, 24, 48, and 72 HAI in four biological replicates. Extraction was performed using 1 mL of freshly prepared sucrose SDS-buffer (100 mM tris-HCl pH 8.0, 30% (*w*/*v*) saccharose, 2% (*w*/*v*) SDS, 0.5% (*v*/*v*) beta-mercaptoethanol, 10 mM EDTA, and protease inhibitor (Merck, Cat. No. 05 892 791 001, Darmstadt, Germany). The samples were incubated at room temperature (RT) for 5 min. Immediately after the extraction, 250 μL of Roti-phenol were added to the samples, which were then vortexed and incubated for 5 min before centrifugation at 20,000× *g* for 5 min at RT. The supernatant was carefully transferred to a new tube. The phenol extraction was repeated. Both phenol fractions were pooled, and counter-extracted with 1 mL of SDS-extraction buffer and centrifuged at 20,000× *g* for 5 min at RT. Protein precipitation was performed by mixing the phenol fraction with 2.5 volumes of 0.1 M ammonium acetate in methanol. After a 16-h incubation period at −20 °C, the samples were centrifuged at 4 °C for 5 min at 5000× *g*. The protein pellets were washed once with 0.1 M ammonium acetate in methanol, and once with ice-cold 70% (*v*/*v*) methanol, centrifuged after each washing step at 4 °C for 2 min at 18,000× *g*, and dried in a SCANVAC Vacuum Concentrater (LaboGene^TM^, Lillerød, Denmark) at room temperature.

Subsequently, proteins were re-suspended in urea buffer (8 M urea, 100 mM ammonium bicarbonate, 5 mM DTT, and protease inhibitor) to measure the protein concentration with a BioRad Bradford Assay with BSA as a standard, prior to protein content normalization. To do so, 200 μg of total protein per sample was first incubated/reduced with dithiothreitol (DTT) at a concentration of 5 mM at 37 °C for 45 min and at 700 rpm in a microcentrifuge. Cysteine residues were alkylated with 55 mM iodoacetamide (IAA) in darkness shaken at 700 rpm at RT for 60 min. Alkylation was stopped by increasing DTT concentration to 10 mM and shaking the samples at 700 rpm in the dark at RT for 15 min. Then, the urea concentration was diluted to 2 M with 100 mM AmBic/10% (*v*/*v*) acetonitrile (ACN). CaCl_2_ was added to a final concentration of 2 mM. Trypsin digestion (Poroszyme immobilized trypsin; 5:100 *v*/*w*) was performed at 37 °C overnight. Peptides were desalted using C18 solid-phase extraction columns (Bond Elut SPEC C18, 96 round-well plate, 15 mg, 1 mL, Agilent Technologies, Santa Clara, CA, USA). After solid-phase extraction, the corresponding eluates were dried in a vacuum concentrator. The peptides were resuspended in 2% (*v*/*v*) acetonitrile, 0.1% (*v*/*v*) formic acid.

The final concentration was measured by a quantitative colorimetric peptide assay (Pierce TM Quantitative Colorimetric Peptide Assay, Thermo Fisher Scientific Inc., Waltham, MA, USA) and 0.5 µg (0.1 µg/µL) were separated on an ES803, EASY-Spray column, 50 cm × 75 µm ID, PepMap RSLC C18, 2 µm (Thermo Fisher Scientific Inc., Waltham, MA, USA). Peptides were eluted using a 90 min linear gradient from 4 to 50% of mobile phase B (mobile phase A: 0.1% [*v*/*v*] FA in water; mobile phase B: 0.1% [*v*/*v*] FA in 80% [*v*/*v*] ACN) with 300 nl/min flow rate generated with an UltiMate 3000 RSLCnano (Thermo Fisher Scientific Inc., Waltham, MA, USA) system. The peptides were measured with an LTQ-Orbitrap Elite (Thermo Fisher Scientific Inc., Waltham, MA, USA) using the following mass analyser settings: ion transfer capillary temperature 250 °C, full scan range 350–1800 m/z, FTMS resolution 60,000. Each FTMS full scan was followed by up to 20 data-dependent (DDA) CID tandem mass spectra (MS/MS spectra) in the linear triple quadrupole (LTQ) mass analyser. Dynamic exclusion was enabled using list size 500 m/z values with exclusion width ± 10 ppm for 30 s. Charge state screening was enabled and unassigned and +1 charged ions were excluded from MS/MS acquisitions. Orbitrap online calibration using internal lock mass calibration on m/z 371.101230 was used.

### 4.5. Data Processing and Protein Identification

The raw data was processed as described in [46] using MaxQuant 1.5. and the Andromeda search algorithm [73,74,75] on the barley Uniprot database (https://www.uniprot.org/, accessed on 21 November 2020). Peptide identification was performed as previously described [76] and statistical analyses were performed with Perseus 1.5. software [77,78]. Unknown proteins were identified by using BLAST at the UniProt homepage searching for the most identical cereal protein. Only those identified proteins were used that were identified in three out of four of the biological replicates over all measured time points. Proteins that were detected only once in two out of four biological replicates or less were dismissed. An average for the missing fourth value was calculated from the three values of the biological replicates. The program GraphPad Prism version 9.0 for Mac, GraphPad Software, San Diego, CA, USA, was used to identify outliers within this dataset (ROUT method, Q = 10%). The deleted values were replaced by the mean value of the three biological replicates. Principal component analysis (PCA) was calculated using parallel analysis within the program GraphPad Prism version 9.0 for Mac, GraphPad Software, San Diego, CA, USA. Proteins were classified into the following groups: “cellular”, “metabolic”, “protein-lipid transport”, “catalytic”, “catabolic”, “unknown”, “defence”, and “seed storage proteins (SSPs)”.

### 4.6. Data Visualization and Statistical Analyses

Data were visualized using GraphPad Prism version 9.0 for Mac, GraphPad Software, San Diego, CA, USA. Statistical analyses were performed with the software Graphpad. Two-way ANOVA followed by Tukey’s multiple comparisons test was performed using GraphPad Prism version 9.0 for Mac, GraphPad Software, San Diego, CA, USA. Hierarchical clustering analysis (HCA) was performed using the online tool https://software.broadinstitute.org/morpheus (accessed on 20 March 2021).

### 4.7. Microscopic Analyses

Eight biological replicates of germinated barley grains including wild-type (GP) and the transgenic TIP3::TIP3-GFP line [30,45] were imaged as described in [30]. Cross sections were prepared after 72 HAI by a razor blade and stained with CellMask^TM^ Green Acting Tracking Stain (1:500 of 1000X stock solution; Invitrogen, Thermo Fisher Scientific Inc., Waltham, MA, USA) for 1 h. Images and time series were taken within 30 min. Sections were analysed with the Leica^®^ SP5 microscope (Leica Microsystems, Wetzlar, Germany) with the following settings: excitation: 503 nm, emission: FITC/GFP filter setting 510–520 nm.

## 5. Conclusions

Here, we took the first steps to reconstitute the germination process #asnearaspossibletonature considering the germination temperature followed by an investigation of the protein dynamics of embryo and endosperm germinated in H_2_O and a high salt solution. Our findings revealed that high salt levels lower the abundance of proteins involved in protein mobilization. The cytoskeleton is affected by salt stress in the embryo as well as in the endosperm during germination. To elucidate the spatiotemporal function of the cytoskeleton during germination, further studies will be required. Finally, long-term studies of grains germinated directly in soil will reveal whether the development of the grains is arrested after the germination delay, indicating that no autotrophic growth will be possible.

## Figures and Tables

**Figure 1 ijms-22-09642-f001:**
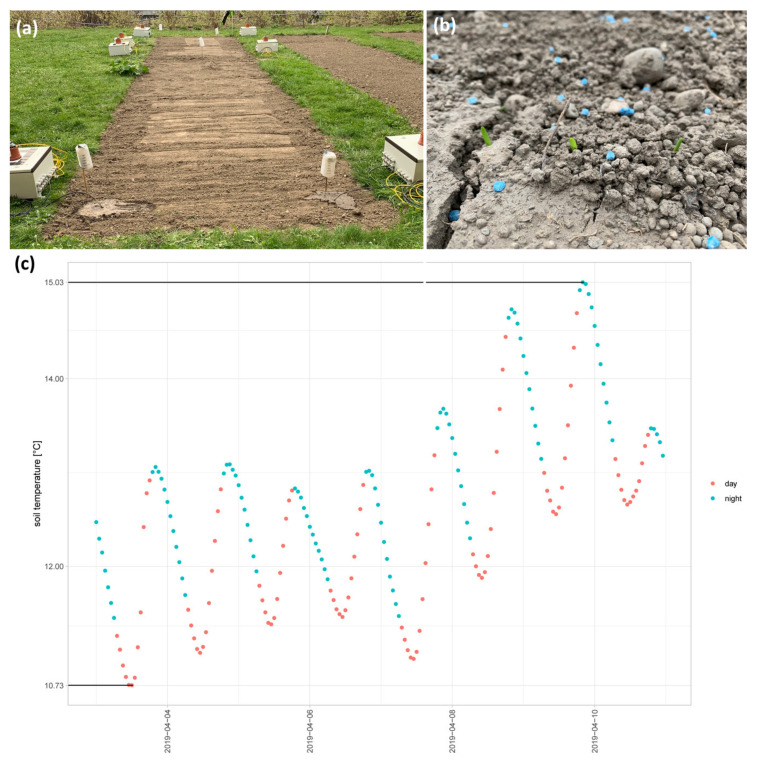
Setup of the field experiment to determine the germination temperature for the barley summer cultivar Golden Promise (GP). (**a**) Seven sensors were used to measure the temperature at different soil depths. (**b**) The shoot was observed seven days after 5000 GP seeds were sown. (**c**) Mean values of data received from five sensors from a soil depth of −20 cm.

**Figure 2 ijms-22-09642-f002:**
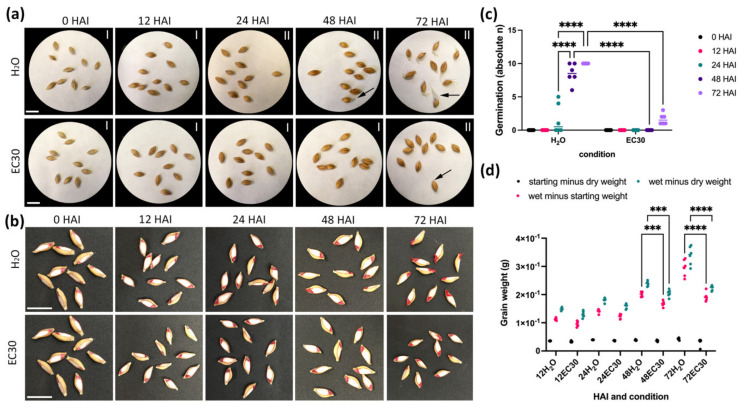
Physiological characterization of germinated GP grains in H_2_O versus EC30 treatment. (**a**,**b**) Physiological and (**c**,**d**) quantitative characterization of germinated grains soaked in H_2_O versus EC30 treatments. The germination phases I and II are indicated describing germination *sensu stricto*. (**a**) Physiological characterization of the germination process in H_2_O versus EC30 treatment. Arrows point to the emerged radicle. (**b**) Viability staining using 1% TCC (2,3,5-Triphenyltetrazolium chloride (TTC) of the germinated grains. (**c**) The germination process was significantly different between 48 and 72 HAI in H_2_O compared to EC30. (**d**) The starting, wet, and dry weight during germination in H_2_O and EC30. The greatest water uptake was at 72 HAI when grains were germinated in H_2_O. Note that in (**b**), the 0 HAI picture is the same for H_2_O and EC30. EC30 = EC30 mS/cm (2 g NaCl in 100 mL tap water). Statistical analyses: two-way ANOVA; significance is indicated with *** (*p*-value ≤ 0.0002) and **** (*p*-values ≤ 0.0001); scale = 1 cm.

**Figure 3 ijms-22-09642-f003:**
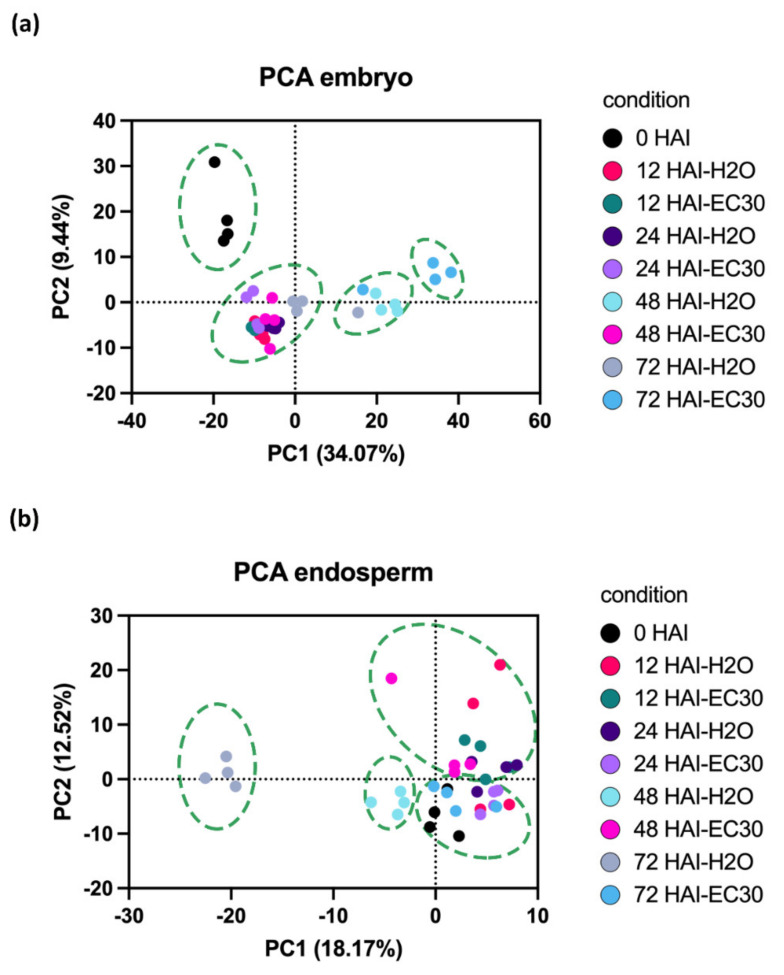
PCA was conducted on protein abundances of the different stages during germination. Each dot corresponds to a single replication (*n* = 4). Protein loadings of the (**a**) embryo and the (**b**) endosperm on PC1 and PC2 were projected in the two-dimensional plan. The different stages were coloured as follows: black circle = 0 HAI, rhodamine = 12 HAI-H_2_O, pine green = 12 HAI-EC30, blue violet = 24 HAI-H_2_O, purple = 24 HAI-EC30, turquoise = 48 HAI-H_2_O, pink = 48 HAI-EC30, grey = 72 HAI-H_2_O, cyan = 72 HAI-EC30. The separated clusters are indicated by the green dashed ellipsoid forms.

**Figure 4 ijms-22-09642-f004:**
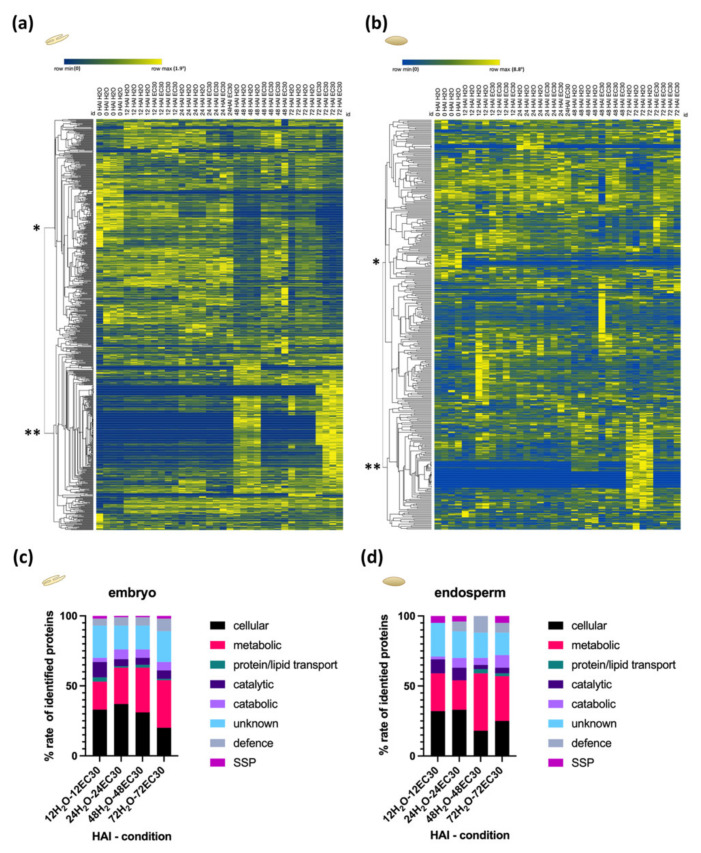
HCA of protein abundances ((labelled-free quantification (LFQ) intensities)) of the different stages during germination and biological classification of identified proteins of the (**a**) embryo and (**b**) the endosperm. * and ** represent the two clusters where the protein abundance is higher and lower, respectively. Blue = low abundance, yellow = high abundance. (**c**,**d**) The biological classification of all significantly differentially expressed proteins during germination in the (**c**) embryo and (**d**) endosperm.

**Figure 5 ijms-22-09642-f005:**
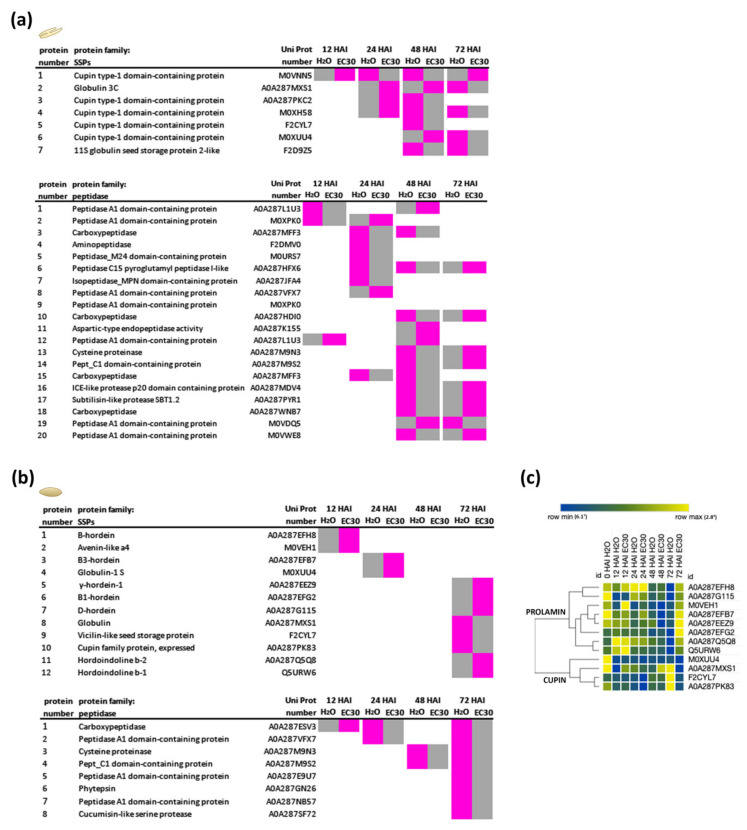
The classification of significantly differently expressed SSPs in the embryo and in the endosperm when germinated in H_2_O and EC30. (**a**) Seven significantly differently expressed SSPs in the embryo. Depending on the point in time, the protein was more (magenta) or less (grey) abundant. All of these proteins belonged to the CUPIN family. A total of 20 peptidases within the embryo were significantly more abundant, mostly at 48 and 72 HAI. (**b**) In the endosperm, the abundance of 12 SSPs was significantly altered. Eight peptidases in the endosperm were most abundant when germinated in H_2_O from 24 HAI on. Magenta = high abundance, grey = low abundance. Statistical analyses: multiple-unpaired t-test. (**c**) HCA analysis of protein abundances (LFQ-intensities) of the SSPs from the PROLAMIN and CUPIN family.

**Figure 6 ijms-22-09642-f006:**
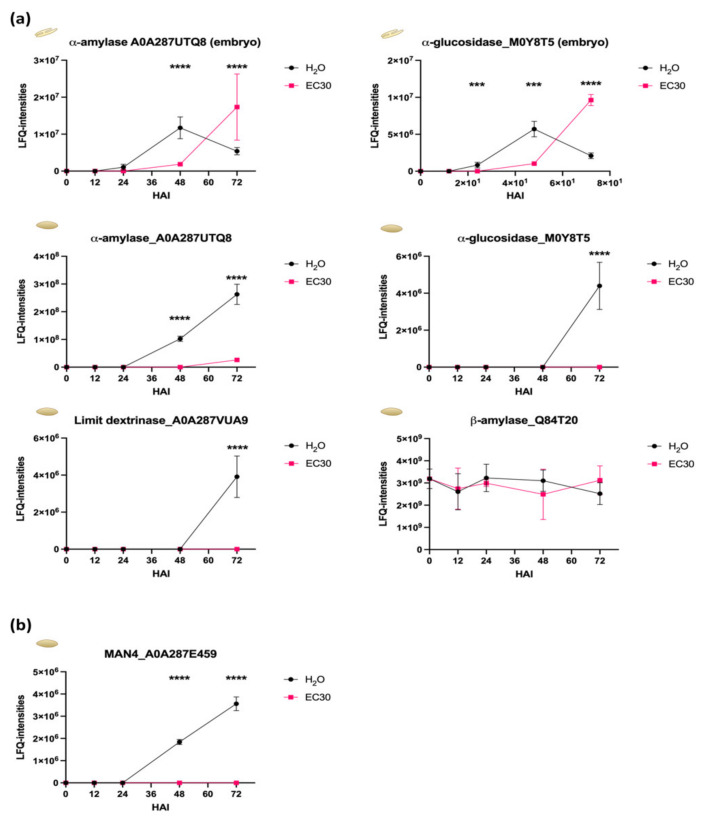
The abundance (LFQ-intensities) of starch and cell wall hydrolysing proteins. (**a**) The abundance of six starch hydrolysing proteins. (**b**) The protein abundance of the cell wall digesting enzyme MAN4 during germination in H_2_O and under EC30 conditions. Statistical analyses: multiple-unpaired t-test; significance is indicated with *** (*p*-value ≤ 0.0002) and **** (*p*-values ≤ 0.0001).

**Figure 7 ijms-22-09642-f007:**
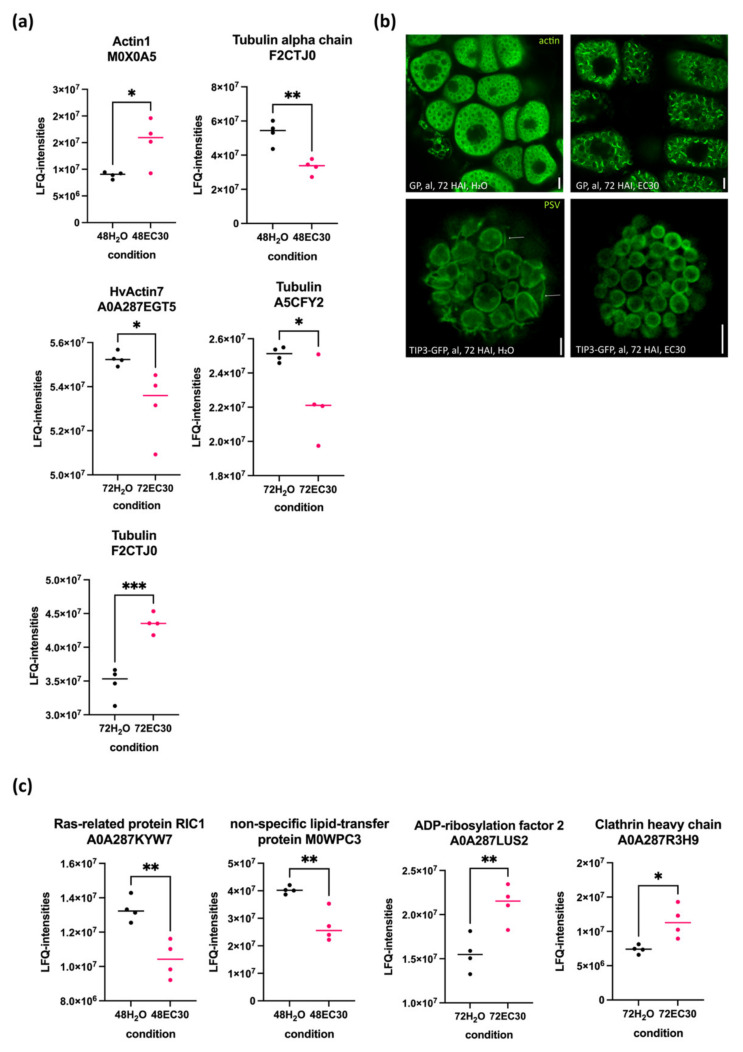
The abundance of proteins (LFQ-intensities) involved in protein trafficking and cytoskeleton-associated proteins in the endosperm when germinated in H_2_O and EC30 at 48 and 72 HAI. (**a**) The abundance of cytoskeleton-associated proteins at 48 and 72 HAI when germinated at 30 EC. (**b**) Confocal micrographs of actin using CellMask^TM^ Green Actin Tracking Stain (1:500 dilution of a 1000X stock solution). Note the tubular structures of PSVs in H_2_O but not in EC30. (**c**) The abundance of four proteins involved in protein trafficking at 48 and 72 HAI. Scale = 5 µm. Statistical analyses: multiple-unpaired t-test; significance is indicated with * (*p*-value ≤ 0.05), ** (*p*-value ≤ 0.01), and *** (*p*-value ≤ 0.0002).

**Figure 8 ijms-22-09642-f008:**
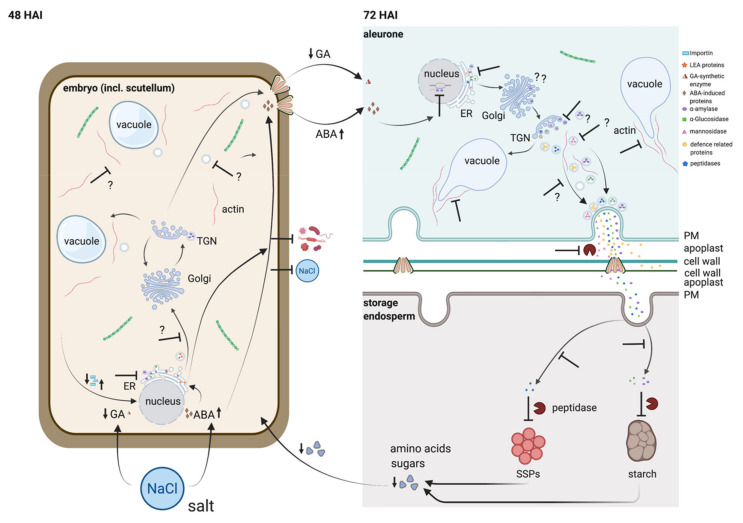
Model for the effect of salt on the cytoskeleton, protein trafficking, and protein mobilization during barley germination. The expression of proteins was most different between H_2_O and EC30 at 48 HAI and 72 HAI in the embryo and endosperm, respectively. We hypothesize that on the one hand, ABA is induced in the embryo by EC30, which inhibits the expression of α-amylase, α-glucosidase, peptidases, mannosidase, and defence-related proteins. On the other hand, the expression of GA_3_-responsive proteins seemed to be negatively affected by the salt treatment. Subsequently, starch and SSPs are less hydrolysed and digested, leading to less mobilization of amino acids and sugars to the embryo. The actin cytoskeleton is altered in the aleurone, and vacuolization is inhibited by salt stress, putatively affecting the vesicular trafficking in the secretory pathway. Arrow up/down: high/low substance level, T-line: inhibition/reduction. Created with BioRender.com.

## Data Availability

Original data will be accessible at the PRIDE Proteomics Identification Database (https://www.ebi.ac.uk/pride/archive).

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
