# Peer review of "Tissue-Specific Proteome and Subcellular Microscopic Analyses Reveal the Effect of High Salt Concentration on Actin Cytoskeleton and Vacuolization in Aleurone Cells during Early Germination of Barley"

_ijms, 2021, doi:10.3390/ijms22179642_

Round 1

Reviewer 1 Report

In the present work, the authors investigate the changes in the proteome in embryo and endosperm of barley seeds during germination under salt stress condition. This part of the manuscript is interesting and clearly provides new data valuable for the research community.

TITLE
The paper title is informative and concise, but strangely worded- what did the authors want to present in the title under the word cytoskeleton?

ABSTRACT
Abstract needs a little improvement. The abstract do not characterize the contents of the paper sufficiently (missing background and aims of work).

MATERIAL AND METHODS
Material and research methods are presented appropriately and clearly. Experimental setup and the description in the methods section are well structured, precise enough, clearly described and the statistical analysis is done alright.

RESULTS
The results obtained in this study are interesting and clearly described. In spite of that I have a few objections against its present form:
-Fig. 1: In my opinion, it is worth indicating at which time points there is phase I and II of imbibition, i.e. germination sensu stricto.
-Figs: Graph titles should not describe the results.
-I cannot fully evaluate the authors' work without the supplementary data that is cited in the text.

DISCUSSION
In general, the discussion of results is correct and sufficient.

LITERATURE
The items of literature included in the paper are rather sufficient and adequate to the subject of the paper.

The text of the manusctipt is not formatted correctly.
Please verify the correctness of the literature and make a linguistic correction of the text by native speaker.

General comments and corrections: I consider the terms germinated embryo and endosperm to be incorrect. I suggest changing to 1) ...embryo and endosperm in germinated seed / ...embryo and endosperm of seed during germination.

lines 41 Correct '...coleorhiza followed by the radicle through the seed coat (germination) [6].' TO '...coleorhiza followed by the radicle through the coleorhiza (germination) [6].' 
lines 44 Correct '...ends with the visible radicle protrusion through the grain coat.' TO '...ends with the visible radicle protrusion through the coleorhiza.'
Comment: In monocot seeds, such as barley, during germination sensu stricto, first the testa ruptures due to growing embryo, followed by radicle protrusion through the coleorhiza.

lines 453 Correct '...70% methanol...' TO '...70% (v/v) methanol...'
lines 467 Correct '...2% Acetonitrile, 0,1% Formic Acid.' TO '...2% (v/v) Acetonitrile, 0,1% (v/v) Formic Acid.' 

Author Response

Point to point reply

We thank the reviewer for reading our manuscript very thoroughly and for all the comments.

In the present work, the authors investigate the changes in the proteome in embryo and endosperm of barley seeds during germination under salt stress condition. This part of the manuscript is interesting and clearly provides new data valuable for the research community.

TITLE
The paper title is informative and concise, but strangely worded- what did the authors want to present in the title under the word cytoskeleton?

We have now revised the title.

ABSTRACT
Abstract needs a little improvement. The abstract do not characterize the contents of the paper sufficiently (missing background and aims of work).

We have nor revised the abstract.

MATERIAL AND METHODS
Material and research methods are presented appropriately and clearly. Experimental setup and the description in the methods section are well structured, precise enough, clearly described and the statistical analysis is done alright.

RESULTS
The results obtained in this study are interesting and clearly described. In spite of that I have a few objections against its present form:

-Fig. 1: In my opinion, it is worth indicating at which time points there is phase I and II of imbibition, i.e. germination sensu stricto.

We have now included the term germination “sensu stricto” in the introduction and indicated the phase I and II in Figure 1.

-Figs: Graph titles should not describe the results.

We have now revised the graph titles and the legends.

-I cannot fully evaluate the authors' work without the supplementary data that is cited in the text.

All supplementary data are provided.

DISCUSSION
In general, the discussion of results is correct and sufficient.

LITERATURE
The items of literature included in the paper are rather sufficient and adequate to the subject of the paper.

The text of the manusctipt is not formatted correctly.
Please verify the correctness of the literature and make a linguistic correction of the text by native speaker.

We have now revised the whole manuscript.

General comments and corrections: I consider the terms germinated embryo and endosperm to be incorrect. I suggest changing to 1) ...embryo and endosperm in germinated seed / ...embryo and endosperm of seed during germination.

As suggested by the reviewer, we have now changed the terms “germinated embryo” and “germinated endosperm” to “embryo” and “endosperm in germinated grains”.

lines 41 Correct '...coleorhiza followed by the radicle through the seed coat (germination) [6].' TO '...coleorhiza followed by the radicle through the coleorhiza (germination) [6].' 

We have now fixed this sentence.

lines 44 Correct '...ends with the visible radicle protrusion through the grain coat.' TO '...ends with the visible radicle protrusion through the coleorhiza.'
Comment: In monocot seeds, such as barley, during germination sensu stricto, first the testa ruptures due to growing embryo, followed by radicle protrusion through the coleorhiza.

We have now fixed this sentence.

lines 453 Correct '...70% methanol...' TO '...70% (v/v) methanol...'

We have now fixed this sentence.

lines 467 Correct '...2% Acetonitrile, 0,1% Formic Acid.' TO '...2% (v/v) Acetonitrile, 0,1% (v/v) Formic Acid.' 

We have fixed this sentence and revised the whole manuscript.

Reviewer 2 Report

Environmental stresses have been potentiated by climate change, having a huge impact on ecosystems and agriculture. In particular, soil salinity, enhanced by drought and erosion effects, is a menace to plant growth / crop production. Plant development needs adequate conditions to grow, especially during germination. Therefore, plants have developed tightly controlled mechanisms to perceive and, consequently, to avoid extreme external conditions. As part of the strategy to avoid salinity, seeds germinate only when external conditions are adequate.

In their manuscript “The cytoskeleton and the vacuolization in aleurone cells are affected by salt during early germination of barley”, Dermendjiev, Schnurer and co-workers investigated the effect of high levels of salinity (i.e. NaCl) on seed proteins during germination. The effect of salt on the molecular response has been studied for some time, with several publications focusing on gene expression (i.e. transcription level) but less on the proteomics level. The topic is relevant because salt can result in lower protein quality, which is particularly important in barley grains and, consequently, in the beverage industry. Thus, barley grains proteomics has been a subject of interest (see review by Bahmani M et al. 2021, DOI: 10.1021/acs.jafc.1c01871). Dermendjiev, Schnurer and co-workers developed an experimental setup closer to field production and performed some physiological analyses. Subsequently, they performed a proteomic analysis of germinating grains exposed to water versus salt condition.

In my opinion, the manuscript is generally well-written and the findings are relevant, but it does contain some issues that should be addressed prior acceptance for publication. I indicate some corrections/clarifications and recommendations in the following sections.

Main concerns

- The title does not reflect the main aspects of the work, and should mention the proteomic analysis of barley grains during germination exposed to salt. Please rephrase the title.

- The first section of the results looks like M&M and contains only supplementary figures. Please rephrase the text and I recommend that some of the supplementary figures to be converted into main figures.

- Certain aspects of protein biosynthesis versus degradation are unclear. It makes sense that SSPs are used as sources during germination (“SSPs must be hydrolyzed by proteases to sustain embryo” (L315); “the degree of the digestion of SSPs increases with the developmental stage during germination in H2O” (L335-336)). However, in the results, many SSPs are accumulating during germination. Could the authors comment on this?

- The staining intensity in Fig 5b must be quantified and statistics calculated to properly support the conclusions.

- I agree with the idea that the experimental design should reflect a more realist and field-like conditions. In line with this, the authors used tap water in the study. Although closer to agronomic reality, it is important to remember that tap water is chemically treated, containing salts. Did the authors analyse the water salt content?

- With the variety of gene expression analysis available, it would be important to include a comparison between the current proteomics analysis with publically available transcriptomic data. This would elevate the quality and robustness of the work.

- The MS contains many mistakes (in particular the M&M and references sections), incorrect writing style (e.g. figure legends) and therefore should be checked carefully. Please see the next list for specific details.

Minor points, corrections and suggestions:

L13: Germination is all about energy -> this statement is not accurate

L17: the most severe factor -> this statement is arguable

L26: are responsible for germination delay -> I do not agree with this statement, the results only suggest this

L76: Barley -> barley

L76: tolerant of alkaline soils -> tolerant to alkaline soils

L91-92: at four different early points -> should be five if include 0 HAI

L92: (HAI) -> (HAI))

L115: tapwater -> tap water

L126: Viability staining -> indicate here the staining method (i.e. TTC staining)

L131: the last graph should be 1d to differentiate from the previous

L134: explain in the legend what EC30 means

L145: EC30, respectively and grains -> EC30, respectively. Grains

L149: eliminate respectively

L152-154: sentence is not clear

L154: Supplement 2b -> figure?

L157: 34,07% -> 34.07%

L159: show -> showed

L159: 18,17% -> 18.17%

L161: Figure Supplement 4b -> Figure Supplement 3b

L166: 1 130 -> remove space

L167: Figure Supplement 3a -> please confirm this reference

L179: HCA -> Hierarchical bi-clustering analysis (HCA)

L180: use another symbol to identify each clustering sections otherwise would confuse with panel letters

L193-194: sentence is not complete -> please check it

L196: SSPs is -> SSPs was

L210: 20 peptidase -> please do not start a sentence with a number

L213: on -> remove

L220: b-amylase -> β-amylase

L221:  table1,2 -> Table 1 and2

L225: Figure 3a -> Figure 4a (actually, each graph should have a letter)

L227-228: Hv should also be in italics

L231: table2 -> Table 2

L250: Acting -> Actin

L258 and 261: Figure S5 -> Figure Supplement 5

L261: are altered -> were altered

L292: do not start a sentence with a number

L293-294: “within” is not correct applied here

L299: the metabolic -> remove the

L299: and SSPs -> and many are SSPs

L303: GA3 -> why mention, they were not included in this section

L312: inhibited or delayed?

L333: Brachypodium distachyon -> italics

L339 and 341: GA3 -> 3 subscript

L345: note, that -> noted that

L353: reveal -> revealed

L360: outstanding -> please check this expression

L363: this section is not supported by the present MS findings

L389: embroy -> embryo

L391: Suppl. -> Supplement

L393-394: sentence need review

L395: many proteins -> many proteins abundance

L417, 425, 457: do not start a sentence with a number

L423: TAP -> tap

L467: Acetronile -> acetronile

L467: 0,1% -> 0.1%

L469: 0,5 and 0,1 -> 0.5 and 0.1

L506: wildtype -> wild type

Figures (in general):

Panel lettering style is different between figures (Fig1 is A and Fig 2 is (a)) -> please uniform.

The legend of a figure should not work as a results section (i.e. Fig 1 L136-137, Fig 4, Fig 5)

Figure 1: last graph should be panel D; germination rate -> the graph does not display a rate but a number of germinated seeds; correct: germination -> Germination; grain -> Grain; 0.0 -> 0; a scale is missing in the pictures; p-values should be < not =; explain the arrows; improve contrast of pictures in panel A (it is hard to seed radicles in white background)

Figure 2: the abundance scale is vague, what are the max and low values?; highlight which categories were statistically enriched

Figure 3: Is the colouring related to a relative value? Otherwise, why is always symmetrical?; aspartic -> Aspartic; scale should numerical have values; legend for panel (c) is missing

Figure 4: explain LFQ in the legend; statistic symbols are missing in the legend; part of the legend is written as it was a result -> please reformulate; each graph should be indicated with different letters; alpha and beta should be replaced with the corresponding symbols

Figure 5: 0.0 -> 0; statistic calculation used is missing; should be > instead of =

Figure 6: different molecules should have different shapes (some colours are alike); the interplay of GA and ABA, as it stands, is confusing

M&M contains several types repeated multiple times:

ml -> mL; units should be separated from numbers (e.g. 2g -> 2 g; 100mL -> 100 mL; 30min -> 30 min; etc) except for ºC (4 ºC -> 4ºC); compounds should not be with capital letter (e.g. Sucrose -> sucrose)

Indicate the plant species/variety in the M&M

References contains several typos and missing information (e.g. DOI), please check this section thoroughly

Revise supplement material order in the text; the authors should uniform the format; please cite supplementary table 3 in the main text; the Figure Supplement 2 should contain the corresponding numbers in the Venn interceptions

Author Response

Point to point reply

We thank the reviewer for reading our manuscript very thoroughly and for all the comments.

Environmental stresses have been potentiated by climate change, having a huge impact on ecosystems and agriculture. In particular, soil salinity, enhanced by drought and erosion effects, is a menace to plant growth / crop production. Plant development needs adequate conditions to grow, especially during germination. Therefore, plants have developed tightly controlled mechanisms to perceive and, consequently, to avoid extreme external conditions. As part of the strategy to avoid salinity, seeds germinate only when external conditions are adequate.

In their manuscript “The cytoskeleton and the vacuolization in aleurone cells are affected by salt during early germination of barley”, Dermendjiev, Schnurer and co-workers investigated the effect of high levels of salinity (i.e. NaCl) on seed proteins during germination. The effect of salt on the molecular response has been studied for some time, with several publications focusing on gene expression (i.e. transcription level) but less on the proteomics level. The topic is relevant because salt can result in lower protein quality, which is particularly important in barley grains and, consequently, in the beverage industry. Thus, barley grains proteomics has been a subject of interest (see review by Bahmani M et al. 2021, DOI: 10.1021/acs.jafc.1c01871). Dermendjiev, Schnurer and co-workers developed an experimental setup closer to field production and performed some physiological analyses. Subsequently, they performed a proteomic analysis of germinating grains exposed to water versus salt condition.

In my opinion, the manuscript is generally well-written and the findings are relevant, but it does contain some issues that should be addressed prior acceptance for publication. I indicate some corrections/clarifications and recommendations in the following sections.

Main concerns 

- The title does not reflect the main aspects of the work, and should mention the proteomic analysis of barley grains during germination exposed to salt. Please rephrase the title.

We have no rephased the title to "Tissue-specific proteome and subcellular microscopic analyses reveal the effect of high salt concentration on actin cytoskeleton and vacuolization in aleurone cells during early germination of barley" to be more specific.

- The first section of the results looks like M&M and contains only supplementary figures. Please rephrase the text and I recommend that some of the supplementary figures to be converted into main figures.

We have revised the first result section and included Supplement Figure 1 and 3 in the main text.

- Certain aspects of protein biosynthesis versus degradation are unclear. It makes sense that SSPs are used as sources during germination (“SSPs must be hydrolyzed by proteases to sustain embryo” (L315); “the degree of the digestion of SSPs increases with the developmental stage during germination in H2O” (L335-336)). However, in the results, many SSPs are accumulating during germination. Could the authors comment on this?

Thanks to the reviewer for pointing this out. We have now discussed the putative major role of the embryo in the germination process facilitating the protein abundance of proteins for protein degradation in the endosperm [1].

- The staining intensity in Fig 5b must be quantified and statistics calculated to properly support the conclusions.

In this case, the staining is to visualize the actin cytoskeleton. The intensity of the observed signal does not reflect the subcellular localization, but the signal itself. We do not quantify with this microscopic analysis the amount of actin but the altered morphological difference between the H2O and EC30 treatment. Within the M&M sections we have described the staining method and the number of biological replicates. We did not observe any tubules in the EC30 treatment.

- I agree with the idea that the experimental design should reflect a more realist and field-like conditions. In line with this, the authors used tap water in the study. Although closer to agronomic reality, it is important to remember that tap water is chemically treated, containing salts. Did the authors analyse the water salt content?

The electric conductivity was measured form tap water as well as from tap water supplied with NaCl. Whereas an electric conductivity of 30 mS/cm was measured in EC30, we measured 0.4 mS/cm in tap water at 22°C. According to https://www.wien.gv.at/wienwasser/qualitaet/ergebnis.html, the tap water of Vienna should have an electric conductivity of 2.5 mS/cm at 16°C, depending on the local area in Vienna. We have now indiated the 0.4 mS/cm for tap water in the M&M part of the manuscript.

- With the variety of gene expression analysis available, it would be important to include a comparison between the current proteomics analysis with publically available transcriptomic data. This would elevate the quality and robustness of the work.

We have now included and discussed available transcriptomic data of germinated barley grains with the focus on GA3 and ABA interplay and the stored and newly produced transcripts in the embryo as well as in the endosperm.

- The MS contains many mistakes (in particular the M&M and references sections), incorrect writing style (e.g. figure legends) and therefore should be checked carefully. Please see the next list for specific details.

Thanks to the reviewer to read our manuscript very thoroughly. We have revised the whole manuscript.

Minor points, corrections and suggestions:

L13: Germination is all about energy -> this statement is not accurate

We have rephrased the introduction sentence.

L17: the most severe factor -> this statement is arguable

We have now changed the sentence to “ “one of the most” to reduce power of the statement. Additionally, the effect of high salt concentration is described in the introduction.

L26: are responsible for germination delay -> I do not agree with this statement, the results only suggest this

We have now rephrased the last sentence.

L76: Barley -> barley

Since we have reduced the introduction according to reviewer 2, this sentence is deleted.

L76: tolerant of alkaline soils -> tolerant to alkaline soils

Since we have reduced the introduction according to reviewer 2, this sentence is deleted.

L91-92: at four different early points -> should be five if include 0 HAI

Thanks to the reviewer, we have changed it to five different early points.

L92: (HAI) -> (HAI))

We have included the second parenthesis.

L115: tapwater -> tap water

We have corrected tapwater to tap water.

L126: Viability staining -> indicate here the staining method (i.e. TTC staining)

We have now included the TTC staining for viability staining.

L131: the last graph should be 1d to differentiate from the previous

We have revised Figure 1.

L134: explain in the legend what EC30 means

We have now explained in the legend that EC30 is EC30 mS/m (2g NaCl in 100ml tap water)

L145: EC30, respectively and grains -> EC30, respectively. Grains

We slit this sentence into to and start with Grains.

L149: eliminate respectively

We have eliminated respectively

L152-154: sentence is not clear

We corrected this mistake.

L154: Supplement 2b -> figure?

As we have moved Supplement Data to the manuscript, the numbers of the Supplement Figures have changed.

L157: 34,07% -> 34.07%

We have changed 34,07% to 34.07%.

L159: show -> showed

We have changed shows to showed.

L159: 18,17% -> 18.17%

We have changed 18,17% to 18.17%.

L161: Figure Supplement 4b -> Figure Supplement 3b

We have corrected this mistake.

L166: 1 130 -> remove space

We have removed the space.

L167: Figure Supplement 3a -> please confirm this reference

As we have moved Supplement Data to the manuscript, the numbers of the Supplement Figures have changed.

L179: HCA -> Hierarchical bi-clustering analysis (HCA)

We have now explained the abbreviation HCA in the text.

L180: use another symbol to identify each clustering sections otherwise would confuse with panel letters

We have revised Figure 2.

L193-194: sentence is not complete -> please check it

We have corrected this mistake.

L196: SSPs is -> SSPs was

We have changed this sentence to past tense.

L210: 20 peptidase -> please do not start a sentence with a number

We have revised this sentence.

L213: on -> remove

We have corrected this.

L220: b-amylase -> β-amylase

We have corrected this.

L221:  table1,2 -> Table 1 and2

We have corrected this.

L225: Figure 3a -> Figure 4a (actually, each graph should have a letter)

We have corrected this.

L227-228: Hv should also be in italics

For the genes, we have written Hv in italics now.

L231: table2 -> Table 2

We have corrected this.

L250: Acting -> Actin

We have corrected this.

L258 and 261: Figure S5 -> Figure Supplement 5

As we have moved Supplement Data to the manuscript, the numbers of the Supplement Figures have changed.

L261: are altered -> were altered

We have corrected this.

L292: do not start a sentence with a number

We have revised this sentence and the whole manuscript.

L293-294: “within” is not correct applied here

We have corrected this.

L299: the metabolic -> remove the

We have deleted the.

L299: and SSPs -> and many are SSPs

We have changed the wording.

L303: GA3 -> why mention, they were not included in this section

We have now included further information on GA3 in this abstract.

L312: inhibited or delayed?

We changed the title of this abstract to “Seed storage proteins in endosperm are less digested in EC30 conditions”

L333: Brachypodium distachyon -> italics

We have corrected this.

L339 and 341: GA3 -> 3 subscript

We have corrected this.

L345: note, that -> noted that

We have revised this sentence.

L353: reveal -> revealed

We have corrected this.

L360: outstanding -> please check this expression

We have revised this sentence.

L363: this section is not supported by the present MS findings

We have reformulated the title of this abstract to make the message more clear.

L389: embroy -> embryo

We have corrected this mistake.

L391: Suppl. -> Supplement

We have corrected this mistake.

L393-394: sentence need review

We have revised this sentence.

L395: many proteins -> many proteins abundance

We have revised this sentence.

L417, 425, 457: do not start a sentence with a number

We have revised these sentences.

L423: TAP -> tap

We have corrected this.

L467: Acetronile -> acetronile

We have corrected this.

L467: 0,1% -> 0.1%

We have corrected this.

L469: 0,5 and 0,1 -> 0.5 and 0.1

We have corrected this.

L506: wildtype -> wild type

We have corrected this.

Figures (in general):

Panel lettering style is different between figures (Fig1 is A and Fig 2 is (a)) -> please uniform.

Thanks to the reviewer. We have corrected this mistake.

The legend of a figure should not work as a results section (i.e. Fig 1 L136-137, Fig 4, Fig 5)

We have revised the figure legends.

Figure 1: last graph should be panel D; germination rate -> the graph does not display a rate but a number of germinated seeds; correct: germination -> Germination; grain -> Grain; 0.0 -> 0; a scale is missing in the pictures; p-values should be < not =; explain the arrows; improve contrast of pictures in panel A (it is hard to seed radicles in white background)

We have now revised Figure 1 including the panel D and scales. We enhanced the contrast and changed the scale.

Figure 2: the abundance scale is vague, what are the max and low values?; highlight which categories were statistically enriched

We have included the LFQ values in the max and low schema. We did not highlight the categories that were significantly different for reasons of the reducing the outline. However, the significantly different proteins are included in supplement table 1 and 2.

Figure 3: Is the colouring related to a relative value? Otherwise, why is always symmetrical?; aspartic -> Aspartic; scale should numerical have values; legend for panel (c) is missing.

The colour explains if the specific protein was more (magenta) or less (grey) abundant at the point in time during germination. We included this information in the legend to improve clarity. We corrected Aspartic and included numerical values in the panel c. C is now also described in the legend.

Figure 4: explain LFQ in the legend; statistic symbols are missing in the legend; part of the legend is written as it was a result -> please reformulate; each graph should be indicated with different letters; alpha and beta should be replaced with the corresponding symbols

We have revised Figure 4. We have explained LFQ intensities now already in Figure 2.

Figure 5: 0.0 -> 0; statistic calculation used is missing; should be > instead of =

We have revised Figure 5 and include the statistic calculation in the legend.

Figure 6: different molecules should have different shapes (some colours are alike); the interplay of GA and ABA, as it stands, is confusing

We have revised Figure 6 (now Figure 8), changed the shape and included further information between the interplay of GA3 and ABA in the legend as well as in the discussion.

M&M contains several types repeated multiple times:

ml -> mL; units should be separated from numbers (e.g. 2g -> 2 g; 100mL -> 100 mL; 30min -> 30 min; etc) except for ºC (4 ºC -> 4ºC); compounds should not be with capital letter (e.g. Sucrose -> sucrose)

We have revised the M&M part.

Indicate the plant species/variety in the M&M

We have included the Golden Promise wild type cultivar in M&Ms.

References contains several typos and missing information (e.g. DOI), please check this section thoroughly

We have edited the reference and included the missing DOI numbers. For 2 publications, no DOI were available.

Revise supplement material order in the text; the authors should uniform the format; please cite supplementary table 3 in the main text; the Figure Supplement 2 should contain the corresponding numbers in the Venn interceptions

We have revised the supplement material and included Figure Supplement 1 and Figure supplement figure 3 in the main text.

  1. Han, C.; He, D.; Li, M.; Yang, P. In-depth proteomic analysis of rice embryo reveals its important roles in seed germination. Plant & cell physiology 2014, 55, 1826-1847, doi:10.1093/pcp/pcu114.

Reviewer 3 Report

The study was focused on investigation of protein dynamics of germinated embryo and endosperm of barley (Hordeum vulgare, L.) grains, subjected to salt stress. The expression of proteins in the embryo as well as in the endosperm appeared to be temporally regulated. Seed storage proteins (SSPs), peptidases and starch digesting enzymes were affected by salt. In addition, the assembly of actin bundles and morphology of protein storage vacuoles (PSVs) in the aleurone layer were altered. The Authors revealed that besides the salt induced protein expression, intracellular trafficking and actin cytoskeleton assembly were responsible for germination delay upon in salt stress.

The paper is interesting. However, I recommend the following improvements of the manuscript in order to increase its scientific soundness and overall readability:

  • The Introduction is too long, hence, I recommend to present it in more concise form.
  • Please, include the name of a statistical test under some figures (e.g. Figure 4).
  • Due to small font and size of axes, some graphs are difficult to read and interpret (e.g. Figure 5 a, 5 c).
  • Minor English language and style changes are required.

Author Response

Point to point reply

We thank the reviewer for reading our manuscript very thoroughly and for all the comments.

The study was focused on investigation of protein dynamics of germinated embryo and endosperm of barley (Hordeum vulgare, L.) grains, subjected to salt stress. The expression of proteins in the embryo as well as in the endosperm appeared to be temporally regulated. Seed storage proteins (SSPs), peptidases and starch digesting enzymes were affected by salt. In addition, the assembly of actin bundles and morphology of protein storage vacuoles (PSVs) in the aleurone layer were altered. The Authors revealed that besides the salt induced protein expression, intracellular trafficking and actin cytoskeleton assembly were responsible for germination delay upon in salt stress.

The paper is interesting. However, I recommend the following improvements of the manuscript in order to increase its scientific soundness and overall readability:

  • The Introduction is too long, hence, I recommend to present it in more concise form.
  • We revised the introduction and tried to be more precise.
  • Please, include the name of a statistical test under some figures (e.g. Figure 4).

We have now indicated the information of statistical tests in the figure legends. 

  • Due to small font and size of axes, some graphs are difficult to read and interpret (e.g. Figure 5 a, 5 c).

We have now changed the arrangement of Figure 5 to increase the size of the diagrams.

  • Minor English language and style changes are required.

We have revised the whole manuscript.

Round 2

Reviewer 1 Report

This paper by Dermendjiev et al. has clearly benefited from the revision, as advised by reviewers.